

# Quantification of In-situ Remediation of Deep Unsaturated Zone and Groundwater

**Ilil Levakov, Zeev Ronen, Tuvia Turkeltaub, Ofer Dahan**

Department of Environmental Hydrology & Microbiology, Zuckerberg Institute for Water Research, Blaustein Institutes for

Desert Research, Ben-Gurion University of the Negev, Sde Boker Campus 8499000, Israel.

*Correspondence to*: Ilil Levakov (ililevakov@gmail.com)

**Abstract.** In-situ bioremediation techniques are cost-effective, environmentally friendly, and sustainable. This study examined a large-scale in-situ treatment of an unsaturated zone and a local groundwater system that were heavily contaminated with perchlorate and other co-contaminants, including nitrate, chlorate, and RDX. Principally, the upper section of the unsaturated

zone was used as a bioreactor for treating the deep unsaturated zone and groundwater. The treatment was based on a cyclic process that included pumping contaminated groundwater, adding an essential electron donor, and injecting the amended water back into the top-soil, which was used as a bioreactor in the treatment process. In the shallow soil, the local bacteria reduced the perchlorate to chloride and water, and the treated water continued to displace the major pollutants from the deep part of the vadose zone, where the biological potential for contaminant degradation is low, towards the water table. The contaminated

leachates were pumped back to the surface with polluted groundwater as part of the cyclic treatment process. Results show that the other co-contaminants, including nitrate, chlorate, and RDX, were removed. Water flow and reactive transport models were calibrated and validated against a time series of the water contents and bromide and perchlorate concentrations that were obtained across the unsaturated zone using the VMS. The calibrated models enabled quantifying the clean-up process and estimating the required time for full perchlorate removal. According to the model's predictions, after 700 days of continuous

operation, all the perchlorate, in a total amount of 7754 kg, would be removed from the unsaturated zone. To obtain full removal, the modelling simulations suggest that the in-situ bioremediation should be implemented for an additional 200 days. Ultimately, we present a low-cost, efficient method for treating perchlorate contamination and potentially that of other pollutants in the subsurface.

## 1    Introduction

Soil pollution by organic and inorganic industrial and agricultural chemicals is a global concern (Juwarkar et al., 2010; Megharaj et al., 2011). Various biological, chemical, and physical soil remediation approaches have been suggested; often, a combination of the methods have offered the most effective treatment (Drenning et al., 2022; Levakov et al., 2020, 2019; Megharaj et al., 2011; Romantschuk et al., 2000). Soil remediation treatments may be applied in situ or ex situ, in which soils

are excavated and treated on site or in designated facilities (Megharaj et al., 2011). In many cases, the in-situ methods are preferable from an ecological perspective (Jørgensen, 2007). Among the in-situ treatment approaches, bioremediation is becoming an increasingly popular alternative to conventional methods (Premnath et al., 2021; Scullion, 2006). Soil bioremediation is mostly based on microbial activity and degradation capabilities. Since natural bioattenuation is relatively



slow, intentional stimulation of chemical degradation is achieved through the addition of water, nutrients, and electron donors

or acceptors (Ritter and Scarborough, 1995).

Perchlorate and other co-contaminants, such as chlorate, nitrate, and RDX, are environmental pollutants mainly released by military industries (Cao et al., 2019; Coates and Achenbach, 2004; Dahan et al., 2017; Gal et al., 2009, 2008). Perchlorate's high solubility and low sorptivity to soil particles facilitate its distribution in the subsurface. In the soil and groundwater, perchlorate remains stable without spontaneous chemical reduction due to its high activation energy (Bardiya and Bae, 2011;

Cao et al., 2019). Due to the large-scale distribution of perchlorate and its co-contaminants in the environment, it is necessary to pursue a suitable, low-cost, and efficient solution to treat and clean both the unsaturated zone and the underlying groundwater.

Perchlorate can be biodegradable under conditions in which it is used as an electron acceptor by native soil and groundwater bacteria (Youngblut et al., 2016). This process is catalysed by (per)chlorate reductase and chlorite dismutase enzymes, which

are responsible for the following two steps: $ClO_4^- \rightarrow ClO_2^- \rightarrow Cl^- + O_2$. Subsequently, the oxygen is also reduced, possibly by the same bacteria, and the final products are chloride and water (Bardiya and Bae, 2011; Coates and Achenbach, 2004). The necessary conditions for reducing perchlorate include the presence of electron donors, a carbon source, molybdenum, and a high soil water content for microbial activity. Moreover, the optimal pH for perchlorate reduction is 7.0, while under higher or lower values, the degradation efficiency is reduced (Shrout and Parkin, 2006; Xu et al., 2003; Zhu et al., 2016). As

thermodynamically more efficient electron acceptors, the presence of oxygen and nitrate in the environment will inhibit perchlorate reduction (Levakov et al., 2020, 2019; N. Liu et al., 2018; Xu et al., 2015; Zhu et al., 2016). Therefore, the absence of nitrate and oxygen is also required for efficient perchlorate reduction. The high reduction potential (E° = 1.287 V) makes perchlorate a favourable electron acceptor under anaerobic conditions (Bardiya and Bae, 2011; Coates and Achenbach, 2004).

For soils and deep unsaturated zones that are polluted by perchlorate, in-situ techniques are used mainly based on

biodegradation by native microbial populations (Bardiya and Bae, 2011; Coates and Achenbach, 2004; Xu et al., 2003). The electron donors are supplied by injecting carbon-rich solutions or gases for the microbial reduction processes at different soil depths. For deep unsaturated zone environments, clean water is injected to displace the pollution to the groundwater from inaccessible layers (Evans et al., 2011; L. Liu et al., 2018; Luciano et al., 2013). Subsequently, the groundwater is pumped for further treatment (Guo et al., 2013; Høisæter et al., 2021). Recently, our group presented a method that is based on the

combination of biodegradation at the upper soil layers and the flushing of contaminants from the deep layers (Avishai et al., 2017; Dahan et al., 2017; Levakov et al., 2019). Analyses of soil samples that were extracted from different depths of the contaminated deep (~40 m) vadose zone site revealed high degradation potential in the shallow soil compared to very low values in the deeper layers (Gal et al., 2008; Sikron, 2013). Moreover, infiltration experiments indicated a major difficulty in applying the electron donor to the deep unsaturated zone (Dahan et al., 2017). Application of the suggested method, the

combination of biodegradation and flushing, was first successfully tested in column and preliminary field experiments (Avishai et al., 2017; Levakov et al., 2019). The main challenge in these experiments was to determine the adequate carbon supply amount to avoid side effects such as acidification or a reduction in biodegradation capabilities (Levakov et al., 2019).

Assessment of the flow and reactive transport processes across the unsaturated zone is a critical requirement for evaluating the treatment's practical applicability. The use of modelling tools enables determining the time scale that is required for the



complete treatment of a contaminated site. The main objective of the current study is to calibrate the unsaturated flow and reactive transport model against long-term physical and chemical field observations obtained from the deep unsaturated zone. These models are implemented to predict and quantify the total time that is required to attain complete remediation of the contaminated unsaturated zone. Ultimately, the efficiency of the long-term treatment of perchlorate and other co-contaminants through combined biodegradation and physical processes was examined.


## 2 Methods

### 2.1 Research site

The experiment was conducted in a former industrial waste pond where high concentrations of perchlorate and other co-contaminants were observed (Gal et al., 2009). Remediation attempts in the area have been conducted since 2009, facing many

difficulties due to the site's unique characteristics (Avishai et al., 2017; Dahan et al., 2017; Gal et al., 2008). The unsaturated zone at the site contains high perchlorate concentrations (up to 30,000 mg l$^{-1}$), located mostly in the deep layers (17–40 m), high salinity (EC > 100 mS cm$^{-1}$), and complex lithology (Figure 1). In the groundwater, the contamination plume is dispersed over an area of several square kilometres (Dahan et al., 2017; Gal et al., 2009). In addition, several co-contaminants, such as HMX, RDX, and TNT, were found in the saturated and unsaturated zones (Bernstein and Ronen, 2012; Sagi-Ben Moshe et al.,

2012). The site's lithological profile is characterized by 40 m of the unsaturated zone containing mostly sand to clayish sand with two layers of high clay content (Figure 1). The first layer, known as NAZAZ, is located at a depth of ~2.5 m, rich with manganese oxides and characterized by low permeability. This layer was excavated and removed in 2010 to improve the infiltration ability in the subsurface (Dahan et al., 2017). The second layer is located at a depth of 13.3 m and contains 25–30% clay content (Figure 1).


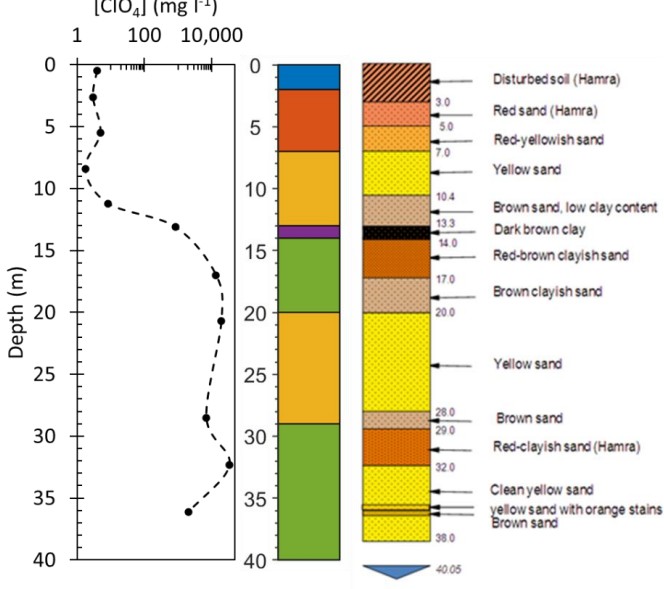



**Figure 1. Perchlorate distribution across the unsaturated zone (left). Model division layers (middle). Lithological profile (right).**

*2.2    Experimental set-up*

The treatment method for remediation of the unsaturated zone and groundwater included cyclic pumping and injection of contaminated groundwater into the topsoil, which was used as the natural bioreactor in the treatment (Figure 2). The experimental site's dimensions were 8 m × 30 m, located inside the main former wastewater pond, on top of a 40-m-deep unsaturated zone. A well (G2), 65-m deep with a perforated section of 25 m below the water table (40–65 m), located in the middle of the treatment zone, was used for daily pumping of contaminated groundwater from beneath the treatment zone. The

daily pumping volume of polluted groundwater was changed during the experiment from 1.7 to 3 m$^3$ per day, according to the percolation conditions and perchlorate degradation efficiency. The treatment efficiency and percolation rate were continuously evaluated, from the variation in sediment moisture and chemical composition across the unsaturated zone as measured by the vadose zone monitoring system (VMS) (Figure 2).

The contaminated groundwater was pumped, using a submersible pump, into a storage tank. A pH buffer, 0.3 mM of potassium

carbonate ($K_2CO_3$) and 1 mM dipotassium phosphate ($K_2PO_4$), was added once a week directly to the storage tank to prevent pH reduction in the treated zone (Figure 2). The water was then injected back into the shallow soil through a subsurface drip irrigation system. During the injection, the water was amended with 1 g l$^{-1}$ ethanol using an automated fertilization system. The ethanol was used as an electron donor and a carbon source in the perchlorate reduction process. The drip irrigation system was spread over the site (240 m$^2$) with a dripper resolution of 30 × 30 cm and a dripping rate of 1.6 l h$^{-1}$ (per dripper). The system

was covered with a polyethylene sheet and 50 cm of local soil on top to prevent oxygen penetration to and evaporation from the treated zone. This experimental stage was conducted for nine months, from October 2019 to June 2020.

A VMS was installed in 2009 at the site, providing real-time monitoring of chemical and hydraulic parameters across the unsaturated zone (Dahan et al., 2017). The system comprises 44 m of a flexible polyurethane sleeve installed diagonally across the vadose zone (35° to the vertical axis) and containing 11 monitoring units (from 0.5 m to 36 m). Each monitoring unit

contains vadose zone sampling ports (VSPs) for frequent sampling of the subsurface porewater and a flexible time-domain reflectometry (FTDR) probe for continuous measurements of the water content. To increase monitoring resolution in the shallow soil, where most of the biodegradation takes place, an additional six monitoring units were installed at depths of 0.05 to 1.5 m. During the experiment, porewater from the soil and the deep unsaturated zone and groundwater were sampled every 6–8 days. All water samples were filtered immediately after sampling and kept at 4 °C before lab analysis.




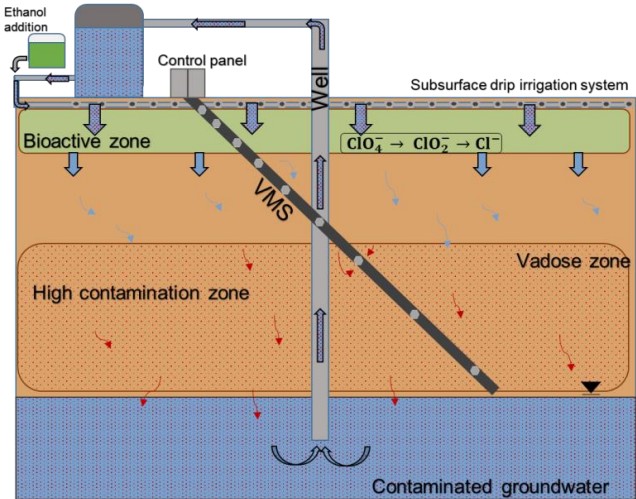

**Figure 2. A schematic illustration of the experimental system (not to scale). The operation starts with pumping contaminated groundwater to an external tank. After the addition of ethanol as the electron donor, the polluted water is injected into the subsurface irrigation system. In the topsoil, the perchlorate**

**is degraded, and the water continues to percolate towards the water table. A vadose-zone monitoring system (VMS) provides real-time monitoring of the degradation and percolation processes.**

*2.3    Tracer experiment*

To assess water flow and conservative transport processes across the unsaturated zone, a tracer experiment that included an injection of bromide as a non-reactive tracer was performed. The bromide was injected in one pulse of 5 m$^{-3}$ with a concentration

of 1785 mg l$^{-1}$. Note that some of the tracer experiment's results were previously presented by Levakov et al. (2019). In the current study, the breakthrough curves of the bromide that were sampled by the VMS were used to calibrate unsaturated water flow and conservative transport models (please see section 2.5).

*2.4    Water chemistry analysis*

Immediately after sampling, pH and EC were measured using pH and EC meters. Perchlorate concentrations were evaluated by an ion-selective perchlorate electrode (Laboratory Perchlorate Ion Electrode, Cole-Parmer). Before each analysis, a calibration curve was prepared with concentrations of 5–1000 mg l$^{-1}$. Samples containing higher concentrations were diluted to the calibration range. All the samples and standards were amended by adding 2% ISA (ionic strength adjustor) at 2 M to

adjust the ionic strength to about 0.12 M. The detection limit of the electrode was 0.1 mg l-1. Bromide concentrations were analysed using the Phenol Red Colorimetric Method (Lepore and Barak, 2009). Nitrate, chlorate, and chloride were analysed by ion chromatography (ICS 5000 Thermo Scientific Dionex). For validation, several samples of bromide and perchlorate concentrations were simultaneously analysed using the ion chromatography system. RDX was analysed by high performance liquid chromatography (HPLC; Agilent 1100 series, Palo Alto, CA, USA).




### 2.5 Water flow and solute transport model

The Richards and the convection–dispersion equations (ADE) were implemented to describe the unsaturated water flow and solute transport in the investigated vadose zone. These equations were numerically solved using the Hydrus 1D code (Šimůnek et al., 2009):

$$\frac{\partial \theta}{\partial t} = \frac{\partial}{\partial z}\left[K(\psi)\left(\frac{\partial \psi}{\partial z} + 1\right)\right], \qquad (1)$$

where z is the vertical direction [L], $\psi$ is the matric head [L], $\theta$ is the volumetric water content [$L^3 L^{-3}$], t is time [T], and K($\psi$) [$L T^{-1}$] is the unsaturated hydraulic conductivity function.

The hydraulic functions of the different soil layers that compose the vadose zone were described by the van Genuchten–Mualem (VGM) formulation (Mualem, 1976; van Genuchten, 1980):

$$S_e = \frac{\theta - \theta_r}{\theta_s - \theta_r} = [1 + (\alpha|\psi|)^n]^{-m}, \qquad (2)$$

$$K(S_e) = K_s S_e^{l}\left[1 - [1 - (S_e)^{1/m}]^m\right]^2, \qquad (3)$$

where Se is the degree of saturation (0 < Se < 1), $\theta s$ [$L^3 L^{-3}$] and $\theta r$ [$L^3 L^{-3}$] are the saturated and residual volumetric soil water contents, respectively, $\alpha$ [$1 L^{-1}$], n [-], and m = (1 – 1/n) are the shape parameters, and Ks [$L T^{-1}$] is the saturated hydraulic conductivity, and the pore connectivity parameter, l, was defined as 0.5.

$$\frac{\partial \theta C_x}{\partial t} = \frac{\partial}{\partial z}\left[\theta D \frac{\partial \theta C_x}{\partial z}\right] - \frac{\partial q C_x}{\partial z} - \mu C_{percolrat}, \qquad (4)$$

where $C_x$ [$M L^{-3}$] is the concentration of bromide or perchlorate in the porewater solution, D [$L^2 T^{-1}$] is the hydrodynamic dispersion coefficient, q [$L T^{-1}$] is the water flux, and $\mu$ [$T^{-1}$] is a first-order perchlorate degradation, which was defined according to previous studies and was further adjusted in the current study (Gal et al., 2009, 2008; Sikron, 2013). The hydrodynamic dispersion coefficient (D) is calculated as follows:

$$D = \lambda v + D_w \tau_w, \qquad (5)$$

where $D_w$ is the molecular diffusion coefficient in free water [$L^2 T^{-1}$], $\tau_w$ is a tortuosity factor in the liquid phase [-], v is the porewater velocity [$L T^{-1}$], and $\lambda$ is the longitudinal dispersivity [L]. The tortuosity factor ($\tau_w$) was determined by Millington and Quirk (1961), and values of $D_w$ were obtained from the literature (Stumm and Morgan., 2012). Note that the $\lambda$ parameter was part of the calibration procedure.

The establishment of the models was conducted in two stages. Firstly, the water flow and the conservative advection-dispersion equation (ADE) models were calibrated against water content measurements and bromide concentrations that were obtained at multiple depths of the vadose zone during the tracer experiment. Subsequently, the first-order coefficients of perchlorate degradation were estimated using data that was collected with the VMS during treatment. Note that although the VMS is installed diagonally and, therefore, each sampling cell represents a separated profile, it is assumed that the unsaturated cross-



section can be presented as a one-dimensional profile. Following this assumption, we applied a 1D flow and transport model
to data that was obtained from a 3D domain.

### 2.6    Model setup and calibration

The simulated soil profile was based on the lithological information that was obtained from sediment sampling during borehole

drilling at the site. Sediment samples were collected at 0.5-m intervals, and the water table was found at a depth of 40 m (Figure
1). Since some of the layers share similar hydraulic characteristics, for the modelling, they were perceived as one layer or
different layers of the same material. For example, red sand (3–5 m) and red-yellowish sand (5–7 m) were considered as one
unit, Layer #2 (3–7 m) in the simulated model profile. Finally, the cross-section included five layers of different materials
(Figure 1).

The upper boundary condition was defined as atmospheric with a surface layer (no runoff; the site is an old evaporation pond),
and the lower boundary condition as deep drainage. The water application was specified according to applied conditions for
both the tracer and the long-term treatment experiments. Note that evaporation is neglected due to the coverage of the irrigation
system and land surface by a polyethylene sheet and 50 cm of soil on top of the sheet. For the water flow model, the initial
conditions were prescribed as water contents that were measured by the VMS. For the ADE model, the upper boundary

condition was prescribed as a concentration flux and zero concentration gradient for the lower boundary condition. The initial
conditions for the bromide tracer were set to zero concentration at the beginning of the model run. For the long-term perchlorate
treatment run, the perchlorate concentrations across the unsaturated profile were prescribed according to porewater
concentrations that were collected through the VMS before the long-term treatment experiment (Figure 1).

The model calibration was conducted in two stages. Firstly, the water flow parameters (e.g., the VGM parameters) and the

conservative transport parameter (i.e., dispersivity; λ) were adjusted according to the bromide concentrations and the water
content measurements (see Levakov et al., 2019) and Supporting Information). The initial hydraulic parameters of the layers
were determined according to the particle size distribution (PSD) using Rosetta (Schaap et al., 2001). Subsequently, the VGM
parameters and the λ parameter were estimated using the "trial and error" approach. In the second stage, the first-order
perchlorate degradation was adjusted according to the long-term perchlorate treatment experiment. To evaluate the overall

predictive performance of the models, the following statistical measures were calculated: (1) root mean square error (RMSE)
and (2) correlation coefficients.

### 2.7    Perchlorate mass balance in the groundwater

The perchlorate concentration changes in the groundwater were calculated through a mass balance on the perchlorate input and

output from a defined volume of the aquifer. Simulated perchlorate that was flushed from the unsaturated zone to the
groundwater was defined as input. The output in the model was the pumped contaminated groundwater from the central well
that is operated daily for recurrent treatment in the site's shallow soil. Since water abstraction by pumping is simultaneously
replenished by the percolating water, it could be considered a mixing problem in a closed domain, considering negligible lateral
flow due to a low regional hydraulic gradient. The calculation for the long-term treatment operation is described in the following

equations:





$$C_{ClO_4}(t) = \frac{M_{GW}(t)}{Vtotal} = \frac{M_{GW}(t-dt) + M_{in}(t) - M_{out}(t)}{Vtotal}, \quad (6)$$

where $C_{ClO_4}(t)$ is the perchlorate concentration in the groundwater at a specific time point (mg l-1), the time division was based on the Hydrus 1D model and set according to the results (day), $M_{GW}(t)$ is the mass of perchlorate in the groundwater (mg), $Vtotal$ is the total volume of the groundwater used for the box model (l), $M_{in}(t)$ is the mass of perchlorate added to the

groundwater during the flushing process (mg), and $M_{out}(t)$ is the mass of perchlorate pumped from the groundwater through the central well (mg).

$$V_{total} = A_{ex} * Z_{box} * WC_s * \frac{1000l}{1m}. \quad (7)$$

The total volume was calculated by multiplying the penetration area, $A_{ex}$ (m$^2$) by the depth of the aquifer section used in the model, $Z_{box}$ (m); $WC_s$ is the water content in the aquifer, estimated as 0.35.

$$M_{GW}(t = 0) = V_{total} * 1000 \frac{mg}{l}. \quad (8)$$

The initial mass of groundwater is the total volume ($V_{total}$) multiplied by the initial perchlorate concentration in the groundwater, which is 1000 mg/l.

$$M_{in}(t) = F(t) * A_{ex} * \frac{1000l}{1m} * C_F(t) * dt. \quad (9)$$

The mass of perchlorate added to the groundwater at a given time is calculated according to the output of the Hydrus 1D model.

F(t) is the flux from the unsaturated zone to the groundwater determined in the model (m/day), and $C_F(t)$ is the concentration of the flux (mg/l).

$$M_{out}(t) = \left( \frac{M_{in}(t) + M_{GW}(t-dt)}{V_{total}} \right) * 1700 \frac{l}{day} * dt \quad (10)$$

The mass of perchlorate pumped out for treatment is calculated by the concentration of the current groundwater (assuming instant mix) multiplied by the daily pumping volume (1700 l per day).

## 3   Results and discussion

### 3.1   Calibration of water flow and conservative transport models

The water flow and conservative transport models were calibrated according to bromide breakthrough curves (Figure 3) and water content measurements (in Supporting Information; Figure 1S) that were obtained at 0.5, 1.5, 2.6, 5.5, and 8.4-m depths. The fitted VGM and longitudinal dispersivity parameters are summarized in Table 1. Note that the adjusted Ks values decay
with depth. This phenomenon of reduction in soil permeability is attributed to soil compaction (Ameli et al., 2016; Turkeltaub et al., 2021). The θs parameters (except for Layer 1) and α parameters did not affect the models' performances, indicating low sensitivity of these parameters (Table 1). The values of the n and θr parameters are mainly related to the texture distribution along the vertical profile presented in Figure 1. Ultimately, the λ of the last two layers are larger by an order of magnitude



compared to the shallower layers (Table 1). It has been previously illustrated that the λ parameter can be depth dependent
(Green et al., 2018; Turkeltaub et al., 2021, 2015; Vanderborght and Vereecken, 2007).

 In simulating bromide concentrations, the water flow and solute transport models achieved RMSE values that are similar to or
lower than the standard deviation (SD) of the observed concentrations (Table 2). According to the statistical measures, the
models show low performances for the bromide concentrations at 2.6 and 5.5-m depths. However, the breakthrough curves at
these depths are not clear (Figure 3). Note that the models succeed in simulating the first arrival time and bromide
concentrations at 8.4 m. An additional comparison was conducted using the water content information from the current field
experiment (Figure 1S). The water content values estimated by the model were compared to the permittivity value measured
by the TDR sensors. Since variations in the salinity profile across the unsaturated zone were very high (Figure 1), the data
obtained from the TDR sensors was presented as permittivity to eliminate the influence of different salinity levels in the
unsaturated zone on the interpreted water content values and to focus on the trends of the water percolation instead of the
absolute water content values.

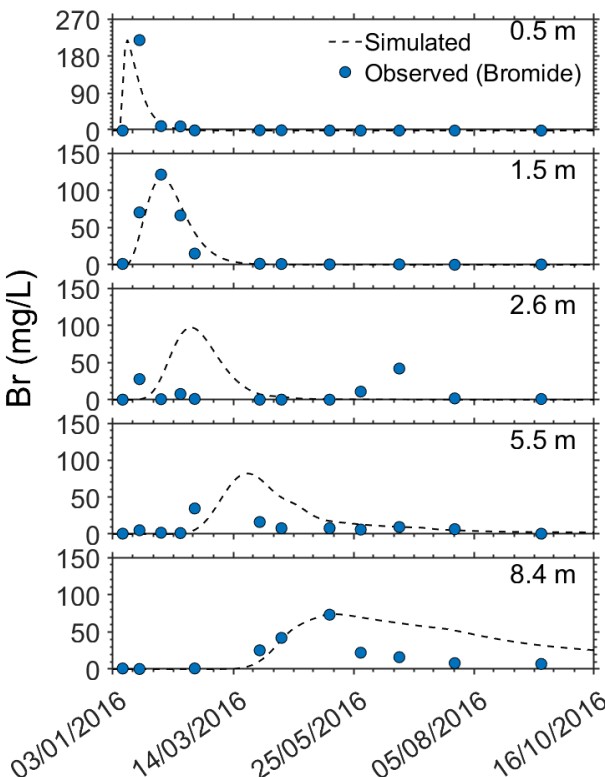

**Figure 3. Observed and simulated bromide concentrations as were sampled from multiple depths of the vadose zone using the VMS.**




Validation of the model in the deepest cross-section was based on the comparison between observed and predicted perchlorate concentrations (Figure 4). At the deepest sampling point (36 m; Figure 4), the initial concentration was relatively low (~2000 mg l⁻¹) compared to the layers above, and therefore, perchlorate concentrations were increased during the experiment to 7000 mg l⁻¹. This is a clear indication of active perchlorate down-leaching to the groundwater. Note that the model simulations

suggested a similar trend (Figure 4). However, at 17-m depth, the perchlorate observations indicated no significant changes in perchlorate concentrations with time, and the model did not predict this behaviour (Figure 4). It appears that the sampling point at 17-m depth is isolated as is also observed by the water content measurements at this point (Figures 1S, 2S). We speculate that the relatively steady perchlorate concentrations throughout the experiment do not represent the processes occurring at 17-m depth. Although a reduction in some of the units' sampling capability in the deep sections of the unsaturated zone was

observed over the years (12 years of continuous operation), the deepest sampling point (36 m) continued to produce water samples that provided important and accurate information on the flow and transport processes occurring in the rest of the unsaturated layers above.

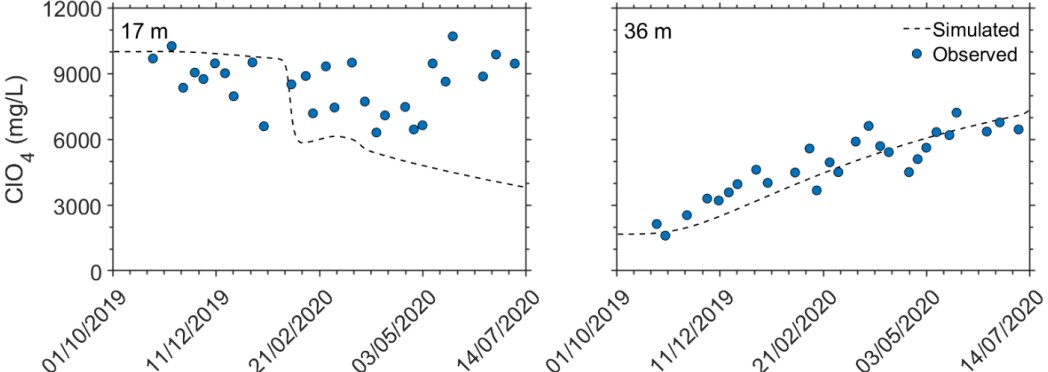

**Figure 4. Comparison of perchlorate concentrations throughout the experiment between the model prediction (dashed line) and the porewater measurements (blue) at 17-m (left panel) and 36-m depths (right panel).**

**Table 1. Summary of the fitted VGM and the longitudinal dispersivity parameters.**

|         | Depth (m) | $\theta_r$ | $\theta_s$ | $\alpha$ (1/cm) | n   | Ks (cm/day) | $\lambda$ |
|---------|-----------|------------|------------|-----------------|-----|-------------|-----------|
| Layer 1 | 0 – 2     | 0.06       | 0.46       | 0.028           | 1.9 | 2000        | 10        |
| Layer 2 | 4 – 7     | 0.04       | 0.37       | 0.028           | 3.3 | 1000        | 10        |
| Layer 3 | 7 – 13    | 0.04       | 0.37       | 0.028           | 3   | 333         | 5         |



| | | | | | | | |
|---|---|---|---|---|---|---|---|
| Layer 4 | 13 – 14 | 0.069 | 0.38 | 0.027 | 1.28 | 13.6 | 5 |
| Layer 5 | 14 – 20 | 0.058 | 0.37 | 0.028 | 1.54 | 70.2 | 5 |
| Layer 6 | 20 – 29 | 0.04 | 0.37 | 0.028 | 3 | 333 | 75 |
| Layer 7 | 29 – 40 | 0.058 | 0.37 | 0.028 | 1.54 | 70.22 | 750 |

**Table 2. Statistics for observed vs. simulated bromide concentrations**

| Depth (m) | RMSE (mg/L) | SD (mg/L) | $R^2$ |
|---|---|---|---|
| 0.5 | 40.2 | 60.2 | 0.5 |
| 1.5 | 13.4 | 41.5 | 0.89 |
| 2.6 | 37.8 | 13 | 0.025 |
| 5.5 | 22.7 | 9.5 | 0.08 |
| 8.4 | 26.3 | 22.4 | 0.4 |

*3.2    Perchlorate biodegradation and transport in the deep vadose zone*

According to the treatment approach, the shallow soil is used as a natural bioreactor in the reduction process. Thus, contaminated groundwater containing 1000 mg l$^{-1}$ of perchlorate is pumped daily from the aquifer and applied to the shallow soil layers after amendment with an electron donor using a subsurface drip irrigation system. The experiment continued for nine months, in which initially, the rate of application was set to 3 m$^3$ per day. However, it appeared that the perchlorate

biodegradation ceased after 126 days (Figure 5). Testing for different technical aspects of the experimental system indicated that the input water flux was too high. Therefore, the rate of water application was changed to 1.7 m$^3$ per day, which substantially improved the perchlorate degradation efficiency.

The initial perchlorate concentrations at most depths were low due to previous treatments (0.5–13 m; Figure 4). In the upper half meter, the concentrations were higher than the groundwater (2000–3000 mg l$^{-1}$), most likely due to perchlorate

accumulation from previous treatments (Figure 4). Once the contaminated groundwater was applied to the topsoil, a decrease in perchlorate concentrations (a flushing process) towards 1000 mg l$^{-1}$ (the groundwater concentration) at 0.05- and 0.5-m depths was observed (Figure 5). The flushing process that was imposed by the intensive water application was expressed as perchlorate breakthrough curves across the vadose zone (Figure 5). This increase in perchlorate concentration is attributed to the early stages when the microbial activity was not yet fully developed. The increase was gradually detected in the soil layers

that contained low perchlorate concentrations (1.3–13 m). Several days after the continued application, evidence for high activity rates of perchlorate-reducing bacteria was observed in the shallow soil layers. As a result, a significant decrease in the





perchlorate concentration was observed at all depths due to microbial reduction in the upper soil layers and a flushing process of the treated water to deeper layers. Similar trends were observed in the previous field experiment (Levakov et al., 2019), while this time, unlike the last experiment, no limiting factor, such as low pH values in the soil, was detected, and the biodegradation efficiency remained consistently high throughout the entire treatment. Perchlorate concentrations at the depth of 0.05 m remained high and unstable due to the short residence time of the polluted groundwater close to the source (Figure 5). Porewater samples from this depth, which are very close to the surface of the soil layers, represent the initial point in the reduction process. As expected, at the depth of 0.5 m, the biodegradation process was sufficient to completely remove the incoming perchlorate below the detection limit. The increase in the perchlorate concentration between 24/12/19 and 2/2/20 was caused by a technical failure that limited the addition of an electron donor to the soil surface. When the system operation was restored, negligible perchlorate concentrations were observed at all depths.

During the treatment, 470 m$^3$ of polluted groundwater was applied to the topsoil with a complete reduction of perchlorate by the surface soil's native microbial population: from 1000 mg l-1 (in the groundwater) to below the detection limit. A mass balance calculation between the applied pollutant and the degradation outcome, as expressed by the concentrations in the unsaturated zone, showed that during this period (nine months), 470 kg of perchlorate was removed from the groundwater, with an average degradation rate of approximately 1.7 kg per day.

To support our interpretation of the observations and to calculate degradation coefficients, the calibrated water flow and transport models were implemented to describe the perchlorate transport in the vadose zone (Figure 5). Technically, the Hydrus 1D does not allow activating the reaction rates at desirable times (e.g., following a maturation period). Therefore, the perchlorate transport model simulation was conducted in two parts. For the first period, the initial conditions of the soil column were according to Figure 1, with no perchlorate degradation. The simulated vadose zone perchlorate concentrations at the end of the first period were implemented as the initial conditions of the second period. Furthermore, Layer 1 from Table 1 was sub-divided into three sublayers to prescribe different degradation rates for different depths (Table 3).

The modelling simulations illustrate that the perchlorate breakthrough curves that were observed along the vadose zone were a consequence of flushing the accumulated perchlorate into the topsoil (Figure 1 and 5). These initial perchlorate concentrations did not experience degradation and showed only conservative transport. Furthermore, according to the model, it takes about 50 days for the microbial activity of perchlorate degradation to be established, in which most of the degradation activity occurs between 0.5 and 1.2-m depths (Table 3). The presence of dissolved oxygen (DO) in the injected water and the minor diffusion of gaseous oxygen reduce part of the perchlorate degradation in the topsoil. Once the oxygen is consumed, perchlorate degradation rates increase. The main uncertainty regarding the fitted degradation coefficients is related to their upper limit. In the current experiment, the groundwater perchlorate concentration was 1000 mg l$^{-1}$, and all the perchlorate was completely degraded.



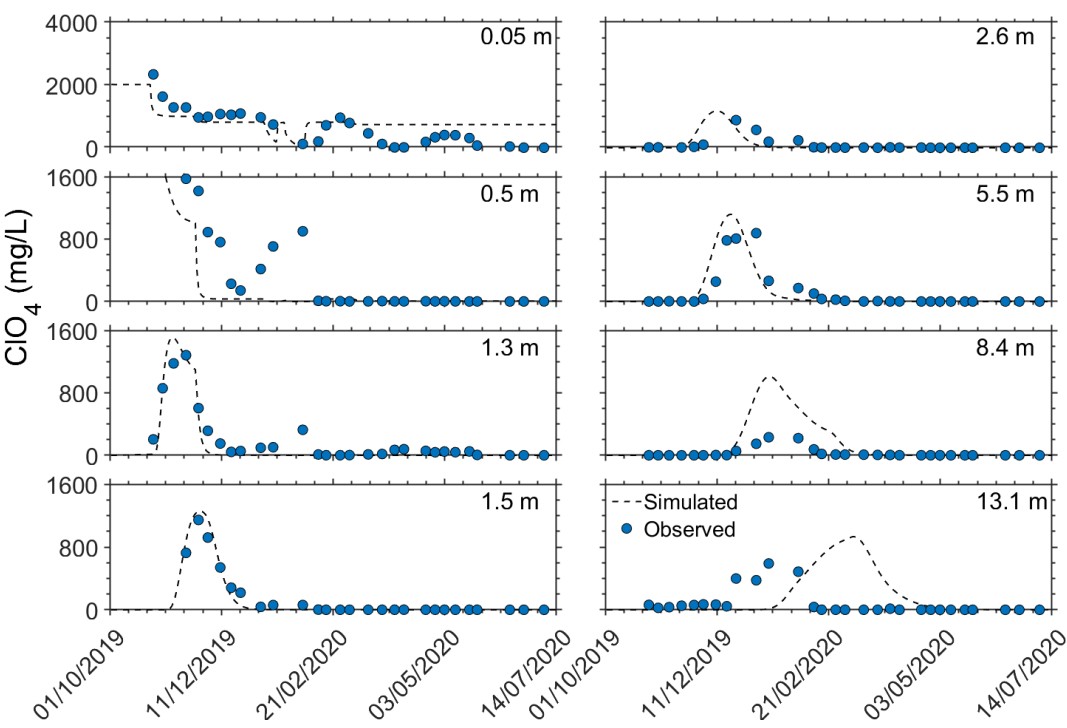

**Figure 5. Observed and simulated perchlorate concentrations as were sampled from multiple depths of the vadose zone using the VMS.**

**Table 3. Calibrated first-reaction coefficients for perchlorate degradation.**

| Depth (m) | $\mu$ (1/day) |
|---|---|
| 0–0.5 | 0.2 |
| 0.5–1.3 | 1 |
| 1.3–40 | 0 |

Further analysis was conducted to test the sensitivity of the calibrated first-reaction coefficients (Table 3) for a range of perchlorate concentrations and water fluxes (Figure 6). The prescribed water flux ranged between 0.7 and 1 cm per day, and the perchlorate concentrations ranged between 1000 and 10000 mg l$^{-1}$. According to the results, the increases in water fluxes required a larger reactive depth compared with the increase in perchlorate concentrations (Figure 6). For a water flux of 0.7 cm per day and 1000 mg l$^{-1}$ perchlorate, the reactive depth was 130 cm. Once the implemented water flux was 1 cm per day, the reactive depth increased to 164 cm. For 0.7 cm per day water flux and 10000 mg l$^{-1}$ perchlorate, the reactive depth was about 150 cm. Thus, the perchlorate treatment was mostly affected by the soil's physical properties. Following previous experience regarding perchlorate treatment (Levakov et al., 2021, 2019), it is reasonable to assume that higher perchlorate concentrations




(> 1000 mg l$^{-1}$ ) would undergo complete degradation. Degradation of less effective electron acceptors, such as sulphate reduction and methanogenesis, as observed in the reactive soil (Levakov et al., 2021, 2020), can indicate a higher potential for

microbial activity. In this case, one would expect larger degradation coefficients than the fitted parameters presented in Table 3. Note that the sensitivity analysis did not account for the effect of large perchlorate concentrations on the microbial activity. This possible effect should be further investigated. Additionally, microbial analysis from previous stages (Levakov et al., 2019) indicated that deeper parts of the vadose zone, up to 2.5-m depth, can contribute to perchlorate degradation and are accounted for as part of the natural bioreactor.


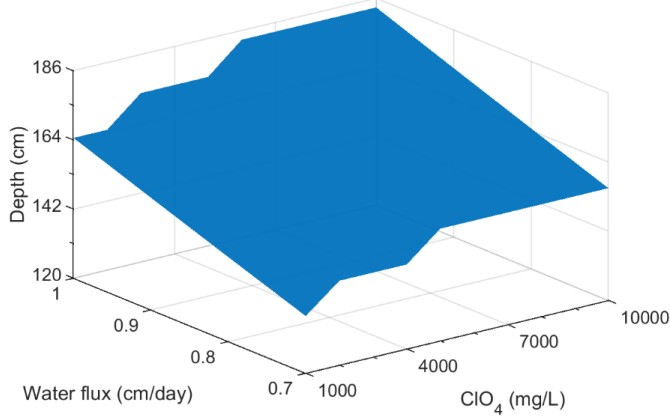

**Figure 6. Three-dimensional depiction of the effect of perchlorate concentrations in the injected contaminated water and the injected water flux rate on the bioactive depth.**


### 3.3    *Perchlorate mass balance for the groundwater*

The mass balance was aimed at calculating the changes in perchlorate concentration in the groundwater for 1000 days according to the long-term simulations. The time was divided according to the results provided by model simulations. The aquifer was described as a closed box with dimensions of 240 m$^2$ × 10 m depth. The perchlorate concentrations, throughout the simulation,

were calculated according to Eqs. 6 to 10 and are presented in Figure 7. The initial perchlorate concentration was 1000 mg l$^{-1}$ as measured from the groundwater samples. During the first 200 days, the concentration moderately decreased due to the low flux from the unsaturated zone at the beginning of the infiltration, with relatively low concentrations at the bottom of the cross-section. As the major mass of perchlorate flushed down the unsaturated zone, as shown at the depth of 36 m (Figure 7a), the concentration in the groundwater increased accordingly. The increase was significant, and the concentration reached 5000 mg

l$^{-1}$ in 300 days. As a result of this expected increase, the estimated daily mass of perchlorate entering the soil should increase accordingly. We assumed that under the optimal conditions, which mainly included maintaining pH ~7, the soil can provide such a high biodegradation rate. According to the current and previous experiments at the site, the percolated water arriving at 0.05-m depth contained negligible perchlorate concentrations (Figure 4). Therefore, the thickness of the bioactive layer used





for full degradation of perchlorate is currently less than 50 cm. Based on the various reduction processes (such as iron and

sulphate reduction, and methanogenesis) occurring in the upper few meters of the soil (Avishai et al., 2017; Levakov et al., 2020, 2019) and the high general microbial activity found at these depths (Levakov et al., 2021), we assume that the biodegradation potential can be increased and the soil can be used as a natural bioreactor for high perchlorate concentrations (as high as 5000 mg l$^{-1}$). After 550 days, when most of the perchlorate was flushed from the vadose zone to the groundwater, as shown in Figure 7, the concentration in the groundwater decreased accordingly. Finally, after 900 days, the mass balance

predicted a full removal of perchlorate from the groundwater. Although the full removal from the unsaturated zone was estimated for 700 days, the operation during the next 200 days is still needed for treating the groundwater. By continuously pumping contaminated groundwater, treating the water in the shallow soil, and displacing the treated water to the water table, the pollution in the groundwater is diluted and replaced by clean water. According to the perchlorate concentrations, the initial mass balance of perchlorate in the groundwater is estimated as 840 kg. During the treatment, 7745 kg of perchlorate will be

displaced from the unsaturated zone to the groundwater. The concentration values of the leachates are predicted to change throughout the treatment duration from 1000–5000 mg l$^{-1}$, and the degradation rate will change accordingly. In conclusion, perchlorate is expected to be fully removed from both the groundwater and the unsaturated zone, providing treatment for a total amount of 8585 kg of perchlorate. It is important to mention that this mass balance does not consider the dispersion and advection in the aquifer, and therefore, a specific model needs to be run after setting the total components of the system, such

as the current surrounding wells and local hydraulic heads. Nevertheless, it provides a rough estimation of the situation in the groundwater.

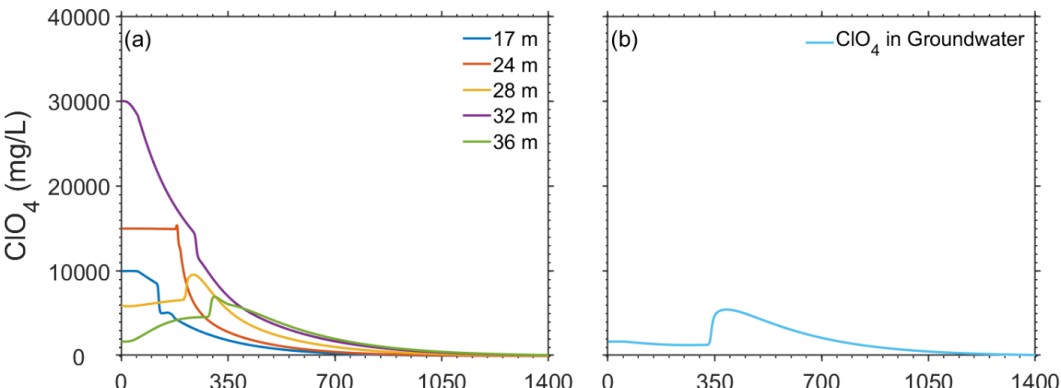

**Figure 7. (a) Estimation of perchlorate concentrations in the unsaturated zone according to the model predictions across**

**three years of system operation. The concentrations are presented at different depths along the deep unsaturated zone. (b) Estimation of perchlorate concentration in the groundwater according to the mass balance based on the model results.**




### 3.4    *Biodegradation of co-contaminants in the unsaturated zone*

In addition to the high concentration of perchlorate in the groundwater, several co-contaminants were found in the polluted groundwater sampled from well G-1, with average concentrations of 200, 100, and 0.5 mg l$^{-1}$ of chlorate, nitrate, and RDX, respectively. Measurements of these three co-contaminants throughout the experiment (Figure 8) demonstrated their significant

decrease during the treatment. The initial concentrations of nitrate and chlorate in the upper 0.5 m were relatively high, with 70–350 mg l$^{-1}$ nitrate and 380–900 mg l$^{-1}$ chlorate. Chlorate, as a potential electron acceptor for most of the (per)chlorate-reducing bacteria, was reduced below the detection limit in the upper 1.5 m a few weeks after the beginning of the treatment (Figure 8). No evidence for the presence of chlorite was observed during the treatment at any depth or in the groundwater (data not shown), meaning the reduction process of perchlorate/chlorate fully occurred, without the presence of by-products. In

addition, denitrification was observed in the reactive shallow soil, which decreased the nitrate concentration to below the detection limit in the upper 1.5 m (Figure 8). Several perchlorate-reducing bacteria were found to be capable of using nitrate as electron acceptors (Youngblut et al., 2016). The process of denitrification is considered thermodynamically preferrable to (per)chlorate reduction (Xu et al., 2003) and, therefore, was observed sooner. Nitrite, as a by-product of the process of denitrification, was not detected throughout the experiment at any depth (data not shown). The increase of chlorate and nitrate

at depths of 2.6 to 13 m was due to the initial application of the polluted groundwater when the microbial population was not yet adapted. The initial pollution front that progressed along the unsaturated zone was partially degraded by the developed microbial populations until 13 m, where no evidence of chlorate and nitrate was observed due to biodegradation and dilution. During the rest of the experiment, while applying high concentrations of chlorate and nitrate to the shallow soil layers, continuous bioreduction was responsible for maintaining the low concentrations of the co-contaminants. In comparison to

nitrate and chlorate, the RDX concentration was relatively low (0.5 mg l$^{-1}$) yet still above the recommended level by two orders of magnitude (Chatterjee et al., 2017). The application of relatively low concentrations to the reactive shallow soil resulted in the rapid removal of RDX from all depths immediately after the operation started (Figure 8). Similar to perchlorate reduction, the RDX concentration remained high at the depth of 0.05 m because of the shallow sampling point, which provides samples of porewater that just arrived at the soil layers. The efficiency of the treatment to remove chlorate, nitrate, and RDX was

demonstrated in the current field experiment.



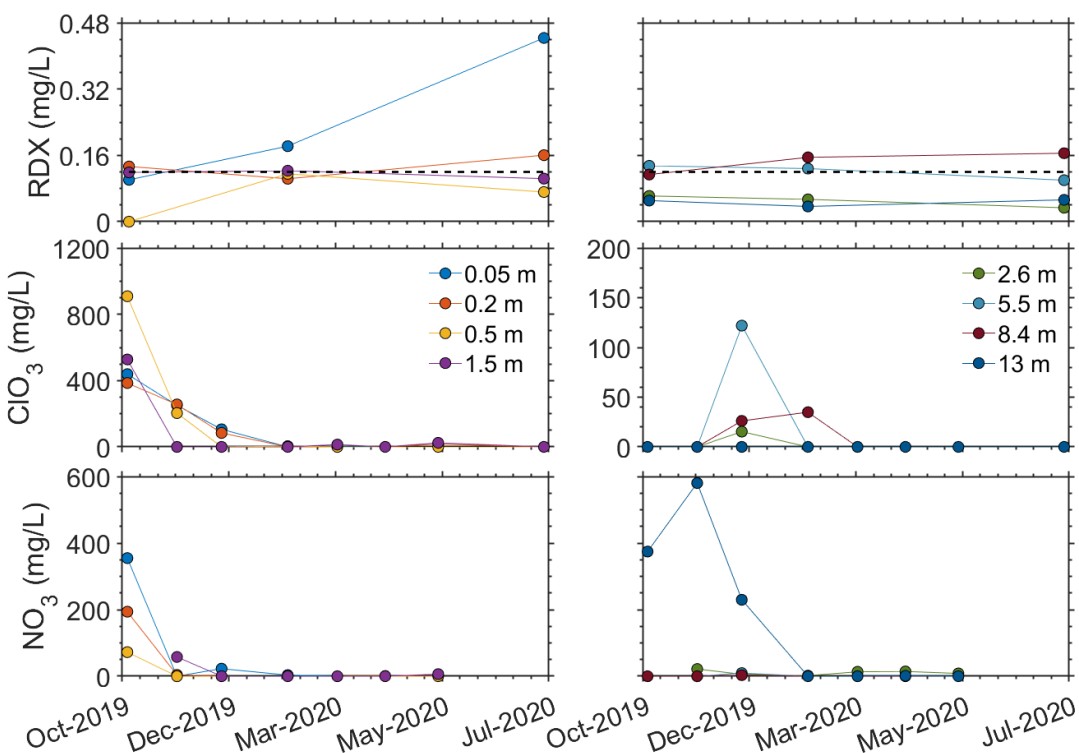

**Figure 8. RDX (top panels), chlorate (middle panels), and nitrate (bottom panels) concentrations that were measured throughout the experiment at different depths of the unsaturated zone (0.05–13 m). The dashed line in the RDX panels represents the detection limit of the analysis.**


## 4    Conclusions

A long-term, large-scale experiment was conducted for testing a combined method for in-situ treatment of perchlorate contamination in the groundwater and the unsaturated zone. The conditions applied during the experiment were implemented according to previous conclusions from a former treatment experiment (Levakov et al., 2019). The major improvements to the

treatment setup were the addition of a buffer to the system to maintain the optimal pH value (~7) for perchlorate degradation and the use of lower concentrations of electron donors to prevent methanogenesis and acidification if excess carbon source is implemented. These additions, which were managed and adjusted continuously according to the actual measured conditions in the unsaturated zone, proved to be efficient for the continuous process of perchlorate degradation in the topsoil. It enabled continuous using of the top-soil as a natural bioreactor for long-term treatment, without the limiting factors associated with pH

decrease. In addition, other co-contaminants were found to be fully reduced during the combined treatment, including chlorate, nitrate, and RDX.

The second part of the treatment process is the flushing of the contamination from the deep unsaturated zone, where the bio-degradation potential of contaminants is low, to the groundwater where it is pumped back for further treatment in the top-soil.



This part was assessed for the first time using the Richards equation and ADE models to estimate the time for full removal of perchlorate from the unsaturated zone. The models were tested and validated according to real measurements from the entire unsaturated zone using temporal variation in water contents, perchlorate concentrations, and tracers migration across the unsaturated zone. According to the model, which was validated by real-time hydraulic and chemical data from the unsaturated zone, the contamination in the unsaturated zone would be fully removed after 700 days of continuous operation, including a daily pumping of 1700 l of polluted groundwater that is reintroduced into the shallow soil after amendments. During the process,

as measured from the model, 7754 kg of perchlorate would be displaced from the unsaturated zone to the groundwater. Subsequently, the contamination would be pumped from the groundwater for treatment in the shallow soil as part of the cyclic process. In order to assess the influence of the treatment on the groundwater, a mass balance model showed that full removal of perchlorate from both the unsaturated zone and groundwater would take 900 days. During this period, the cyclic process would include pumping of contaminated groundwater and biodegradation of the perchlorate and other contaminants in the

upper soil. Hence, according to this study, operating the current treatment method for 900 days (~2.5 years), which requires minimal costs, would provide an efficient solution for the perchlorate contamination in the unsaturated zone and the groundwater. In this case study, the estimated removal mass of perchlorate from both the unsaturated zone and groundwater, underlying an area of 240 m$^2$, is ~8585 kg.

**Author contribution**

Ofer Dahan and Zeev Ronen: conceptualization, investigation, methodology, validation, supervision, writing: review and editing. Tuvia Turkeltaub: software, writing: review and editing. Ilil Levakov: conceptualization, formal analysis, investigation, visualization, writing: original draft preparation.

**Competing interests**

The authors declare that they have no conflict of interest.

**Acknowledgements**

The authors wish to express their gratitude to Michael Kugel and Amos Russak for their technical assistance with the
experiments and analysis. The work was funded in part by the Israel Water Authority (scholarship #8778381), Netzer Hasharon LTD., and by the Israel Science and Technology Ministry (grant 3-17408).





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
