# Peer review of "Quantification of In-situ Remediation of Deep Unsaturated Zone and Groundwater"

_EGUsphere, 2022_

## Referee Comment (RC1)

The authors present a novel in-situ, bioremediation method for the treatment of polluted waters from the unsaturated zone and from the groundwater. The method relies on a cyclic process in which contaminated groundwater is pumped to the surface and subsequently, it is injected back into the topsoil (after the addition of an electron donor). The topsoil area acts like a bioreactor in which contaminants are removed by microbial degradation. Cleaned water is then propagating through the unsaturated zone to greater depths and pushing down further contaminated water fractions to the groundwater where it is then again pumped back to the surface, closing the cyclic process. In this way, all contaminants are gradually removed.

The authors present experimental results on the removing of perchlorate and other contaminants at a field site over several months. They calibrate and validate the HYDRUS 1D model on experimental data (i) to capture the observed breakthrough behaviour of perchlorate over the vertical extent of the unsaturated zone, (ii) to quantify the perchlorate breakthrough into the groundwater aquifer and (iii) to predict the time scale for a complete removal of perchlorate and all other contaminants at the specific field site.

I think that such technical methods for removing contaminants out of the subsoil are a crucial tool for protecting water resources and ecosystem. Hence, the present study is relevant for the readers of this journal. In general, I could quite easily follow the descriptions and common thread of the study as the structure and build-up of the text was clear to me. However, despite the relevance of the topic and the good readability of the study, I would recommend some moderate to major revisions due to the following general and specific comments.

**General comments**

1. For me it remains quite unclear which of the presented experimental results/data have been published before and which are completely new in this study. The reader can get the impression that the former is true for all observed data and that "only" the simulation part is new (until reading the Conclusions). At the beginning of the study, please make explicitly clear in a short passage which experimental results are new and which have been published before, and maybe also stress the differences of the current experiment setups compared to the previous experiments at that site.

2. For the simulation part, it would be interesting to see uncertainty ranges and sensitivities of the simulation results to different soil hydraulic parameterizations (at least for the calibration), as models like HYDRUS have several degrees of freedom and are quite sensitive to the soil hydraulic functions.

3. In the discussion, the authors may additionally comment on:

    (i) The influence of preferential leaching (e.g. in macropores) of contaminants at that site. Don't you have any evidence for preferential leaching and a bypassing of the bioreactor zone through heterogeneous soil structures, because you obviously use an one-porosity approach for the HYDRUS simulations assuming well-mixed conditions? I think especially for the presented in-situ remediation method with biodegradation in the shallow unsaturated soil zone the assessment of the influence of preferential leaching is crucial, also for the general transferability and applicability of the method at other sites.

(ii)    General differences of the presented method to other, commonly used methods in terms of, e.g. the scope, costs, environmental sustainability, risks and challenges.

**Specific comments**

1. P. 1, abstract: You use some abbreviations directly at the beginning of your study without giving their entire name once when using the terms for the first time. Please also check this issue throughout the study as it occurs more often (e.g. in the methods section).

2. P. 1, l. 32: I was wondering if there is one classical method that is most commonly used for remediation to which the presented in-situ bioremediation method and the specific perchlorate results of this study can be directly compared (cf. my last general comment).

3. P. 2, l. 36-42: How relevant is the contamination of ecosystems by perchlorate. Do you have any specific numbers on the magnitude of perchlorate contamination on a global or national scale?

4. P.3, Figure 1: The quality of the legend picture is poor and hardy readable. Please try to increase the readability. Also, the left part of the legend is not further defined here but is only described later in the text. It would be easier for the reader if these 5 different soil layers were already properly defined here in Figure 1.

5. P. 4, l. 115: Which suction pressure/head was used to sample the pore water in different depths? Was it a constant pressure or variable depending on the soil water content? Please provide more information on that.

6. P. 5, Figure 2: I really like this schematic sketch, it provides a good overview.

7. P. 6, l. 147: In this case it must be "advection-dispersion-equation (ADE)".

8. P. 6, l. 165: How was $v$ determined? Measured, calibrated? In general, what are the used values for $v, D, \tau_w$? Maybe you can show a Table with all parameters values used for the simulations, at least in the Appendix.

9. P. 7, l. 196: Can you please provide some more information about the determination of hydraulic properties of the different soil layers? In the previous experiments at this site over the last years, were there no actual measurements of the hydraulic properties (and the generation of soil water retention and hydraulic conductivity curves) of soil samples in different depths?

10. P. 7, l. 207: This would imply a steady-state condition. Is there any experimental evidence for this assumption? And in the next line, how low is this regional gradient that reasons the closed domain assumption?

11. P. 8, l. 217: In line with my specific comment #8, can you please provide an overview of the used parameter values of $A_{ex}, Z_{box}, V_{total}$?

12. P. 8, l. 221: I think you mean "The initial mass of perchlorate is…".

13. P. 8, Eqs. 7 and 8: The unit conversion factor must be "1000 L / 1 m³" for consistency.

14. P. 9, l. 239: Can you here please give a short explanation or suggestion why the diffusivity in the last two layers is so high compared to the other layers above?

15. P. 10, l. 263: Why even showing and using the observed values in 17 m depth for simulation, when you think that this sampling point does not generate reliable measurements and not capturing the real conditions in this depth?

16. P. 10, Table 1: The unit of $\lambda$ is missing.

17. P. 12, l. 310-313: This explanation is not completely clear to me. Can you please provide some more information on this modelling procedure in two phases? How long are the respective phases? And how do you subdivide Layer 1 from Table 1 (0-2m) into the three sublayers of Table 3 between 0-40 m?

18. P. 13, Table 3: Do you assume $\mu$ as the first-order degradation rate coefficient for the water phase? Thus, do you only assume degradation taking place in the water phase, as you do not say anything about parameterizing sorption? Does perchlorate not adsorb at all to soil particles?

19. P. 14, l. 363: I cannot find a depth of 0.05 m in Figure 4. Please revise.

20. P. 15, Figure 7: Labels of x-axis are missing.

21. P. 18, l. 446: What does "minimal costs" actually mean? What would a continuous treatment over 900 days cost and what is the difference to common methods?

---

## Author Comment (AC1)

**General comments**

**Comment 1:** For me it remains quite unclear which of the presented experimental results/data have been published before and which are completely new in this study. The reader can get the impression that the former is true for all observed data and that "only" the simulation part is new (until reading the Conclusions). At the beginning of the study, please make explicitly clear in a short passage which experimental results are new and which have been published before, and maybe also stress the differences of the current experiment setups compared to the previous experiments at that site.

**Reply to comment 1:**

The comment is accepted, and the manuscript was revised accordingly. (The only results that were previously published are the trace experiment).

Lines 104: " The study includes a new treatment experiment conducted for nine months from October 2019 to June 2020." Line 131: " The current remediation experiment is presented for the first time "

Lines 143-145:

"In addition to the new result from the long-term experiment (section 2.2), former data from a trace experiment was used for the assessment of the water flow in the unsaturated zone. This experiment was conducted in 2015 in the same research field and previously presented by Levakov et al. (2019)."

The differences between the current and former experiments are mentioned in the conclusions section in lines 425-429. Nevertheless, an additional explanation was added to the manuscript: lines 131-134: " The current remediation experiment is presented for the first time, and the applied conditions in it were based on previous conclusions from the research site. The recent improvements to the experiment setup included the addition of a buffer to the system and the use of lower concentrations of electron donors in order to maintain optimal pH values for the degradation process."

**Comment 2:** For the simulation part, it would be interesting to see uncertainty ranges and sensitivities of the simulation results to different soil hydraulic parameterizations (at least for the calibration), as models like HYDRUS have several degrees of freedom and are quite sensitive to the soil hydraulic functions.

**Reply to comment 2:** We accepted the reviewer's suggestion, and the Morris method sensitivity analysis has been applied (Morris, 1991). In this method, the parameters are modified one-at-a-time (OAT) and sensitivity is estimated as the partial derivative of the change in model output for a given change in a single input parameter (Perzan et al., 2021). We used the SAFE Matlab code provided by Pianosi et al. (2015). Furthermore, the sensitive parameters were calibrated following the method presented by Perzan et al. (2021), where uncertainty and optimal values are calculated according to the behavioral simulations (by achieving evaluation goals). Note that the 'Methods' and the 'Results and 'discussion' sections have been revised (Lines 214-250;282-316; Tables 1 and 2; Figures 3-6).

**Comment 3:** In the discussion, the authors may additionally comment on:

(i) The influence of preferential leaching (e.g. in macropores) of contaminants at that site. Don't you have any evidence for preferential leaching and a bypassing of the bioreactor zone through heterogeneous soil structures, because you obviously use an one-porosity approach for the HYDRUS simulations assuming well-mixed conditions? I think especially for the presented in-situ remediation method with biodegradation in the shallow unsaturated soil zone the assessment of the influence of preferential leaching is crucial, also for the general transferability and applicability of the method at other sites.

**Reply to comment 3(i):** Preferential flow in the unsaturated zone is expected. Moreover, our monitoring technology which measures water flow and solute transport in the unsaturated zone provides a clear indication for preferential flow as it provides information from multiple independent monitoring points across the unsaturated zone, where each point is located under a different unsaturated column. Quantifying the extent of the preferential flow both as reflected from the field

measurement and through the model is not straightforward. Yet, in this study, which was conducted in unconsolidated sandy and clay sediment, we could clearly see that the infiltrating water does not have a significant bypass of the bioactive zone. The indications come from the fact that multiple monitoring points at different depths under different soil profiles where all show reasonably coherent results. This enabled us to simplify the model while accounting for single soil properties in the different layers.

(ii) General differences of the presented method to other, commonly used methods in terms of, e.g. the scope, costs, environmental sustainability, risks and challenges.
**Reply to comment 3(ii):**
The comment is accepted, and the manuscript was revised accordingly.
The common methods for perchlorate treatment were elaborated in the introduction: lines 58-68:
"Perchlorate contaminations are treated around the world according to the nature of the contaminated site. In the case of polluted soil and deep unsaturated zone, in-situ techniques are used mainly based on biodegradation by native microbial populations (Bardiya and Bae, 2011; Coates and Achenbach, 2004; Xu et al., 2003). For example, injections of carbon-rich solutions (Frankel and Owsianiak, 2005) or gases (Cai et al., 2010; Evans et al., 2011) supplied electron donors for the microbial reduction processes directly to the soil at different depths up to 10m. As opposed to those methods, the current research site is characterized with a deep unsaturated cross-section (40m) while the major mass of perchlorate is located in the deep layers (17-36 m) (Dahan et al., 2017; Gal et al., 2009; Levakov et al., 2019). For pollution in the deep unsaturated zone, flushing is a common way to treat deep and inaccessible layers by injecting clean water to the site, displacing the pollutions to the groundwater, and collecting it back to the surface for further treatment (L. Liu et al., 2018; Luciano et al., 2013; Wellman et al., 2011). Usually, the subsequent treatment includes ex-situ methods outside the contaminated site (Guo et al., 2013; Høisæter et al., 2021)"
And the comparison to our technique was added to the conclusions: lines 528-536:
"Usually, different in-situ biodegradation techniques are used for contaminated soils up to a deep of 10m, with lower perchlorate concentrations (Bardiya and Bae, 2011; Coates and Achenbach, 2004; Xu et al., 2003, Frankel and Owsianiak, 2005, Cai et al., 2010; Evans et al., 2011). An additional approach is an excavation of the contaminated soil followed by an ex-situ treatment outside the site. For deeper contamination, the common treatment approach involves an injection of clean water, displacing the pollution towards the water table, and collecting it back to the surface for further treatment ex-situly. All ex-situ approaches require relatively high costs and complicated operations during the transport of the contaminants and the establishment of the treatment facilities. Unlike those methods, the proposed approach treats the contamination in-situly and simultaneously for the soil, the unsaturated zone and the groundwater with extremely high perchlorate concentrations in a relatively simple and affordable manner."

**Specific comments**
**1**. P. 1, abstract: You use some abbreviations directly at the beginning of your study without giving their entire name once when using the terms for the first time. Please also check this issue throughout the study as it occurs more often (e.g. in the methods section).
**Reply:** The comment is accepted, and the manuscript was revised accordingly. Lines 9,16,18,35, and 92.

**2**. P. 1, l. 32: I was wondering if there is one classical method that is most commonly used for remediation to which the presented in-situ bioremediation method and the specific perchlorate results of this study can be directly compared (cf. my last general comment).
**Reply:** The comment is accepted, see reply to comment 3(ii).

**3.** P. 2, l. 36-42: How relevant is the contamination of ecosystems by perchlorate. Do you have any specific numbers on the magnitude of perchlorate contamination on a global or national scale?

**Reply:** The comment is accepted, and the introduction section was revised accordingly. Lines 39-41:
" Since the late 1990s, the recognition of perchlorate contamination was grown worldwide, especially in the USA, Canada, China, and west Europe with thousands of studies examining perchlorate contaminations in different water bodies, soils, and food products (Cao et al., 2019). "

**4**. P.3, Figure 1: The quality of the legend picture is poor and hardy readable. Please try to increase the readability. Also, the left part of the legend is not further defined here but is only described later in the text. It would be easier for the reader if these 5 different soil layers were already properly defined here in Figure 1.
**Reply:** the comment is accepted. The figure was updated and the caption was revised. Lines 101-102:
"**Figure 1. Perchlorate distribution across the unsaturated zone (left). Lithological profile (right)***, and division of the profile into five different layers as used for the Hydrus 1D model (middle).***"

**5**. P. 4, l. 115: Which suction pressure/head was used to sample the pore water in different depths? Was it a constant pressure or variable depending on the soil water content? Please provide more information on that.
**Reply:** The technical information on the vadose zone monitoring system was published previously with an accurate description of the system's operation. In order to keep the manuscript relevant and concise we try to avoid excessive details. Nevertheless, we added a reference for more detailed information. Lines 128-129:
" For more detailed information on the vadose zone monitoring system see Dahan et al., 2009. "

**6**. P. 5, Figure 2: I really like this schematic sketch, it provides a good overview.
**Reply:** Thank you

**7**. P. 6, l. 147: In this case it must be "advection-dispersion-equation (ADE)".
**Reply:**  The comment is accepted, and the term was revised. Line 163.

**8**. P. 6, l. 165: How was *v* determined? Measured, calibrated? In general, what are the used values for *v, D, τw*? Maybe you can show a Table with all parameters values used for the simulations, at least in the Appendix.
**Reply:** The *v* is the porewater velocity and it is assumed to be equal to the ratio between the calculated Darcian flux (calculated by the Richards equation) and the water content. Note that these values are calculated with the numerical model integrated in Hydrus 1D code (we add an explanation to the text, lines 185-186). The calculations of the Richards equation and the Advection-Dispersion equations are coupled. Similarly goes for the tortuosity ($\tau_w = \frac{\theta^{7/3}}{\theta_s^2}$), which changes according to the water state of the soil (see line 183). As described in equation 5 (line 181), the hydrodynamic dispersion coefficient (D) is a function of Darcian flux (or porewater velocity) (Richards equation), tortuosity, longitudinal dispersivity ($\lambda$), which is handled as a fitting parameter (lines 184-185) and $D_w$, which is the diffusion coefficient that accounts for the Brownian movement. The $D_w$ was obtained from literature, $1.55 \times 10^{-4}$ m$^2$/day (lines 184-186).

**9**. P. 7, l. 196: Can you please provide some more information about the determination of hydraulic properties of the different soil layers? In the previous experiments at this site over the last years, were there no actual measurements of the hydraulic properties (and the generation of soil water retention and hydraulic conductivity curves) of soil samples in different depths?
**Reply:** We agree with the reviewer's comment and further explanation regarding the determination of the hydraulic parameters is provided in the text (Lines 214-250).

**10**. P. 7, l. 207: This would imply a steady-state condition. Is there any experimental evidence for this assumption? And in the next line, how low is this regional gradient that reasons the closed domain assumption?

**Reply:** The situation in the groundwater was simplified by a steady-state condition in order to provide a picture regarding the influence of the treatment on the groundwater pollution. Although there is horizontal flow in the aquifer, it considers low especially compared to the pumping/injecting rate from/to the aquifer.  Moreover, we clarified this inaccuracy in lines 391-394:

" It is important to mention that this mass balance does not consider the dispersion and advection in the aquifer, and therefore, a specific model needs to be run after setting the total components of the system, such as the current surrounding wells and local hydraulic heads. Nevertheless, it provides a rough estimation of the situation in the groundwater."

Additional clarification was added to the methods section. Lines 253-254:

" The calculation was simplified by a steady-state condition in order to provide a rough estimation for the contamination in the groundwater "

**11**. P. 8, l. 217: In line with my specific comment #8, can you please provide an overview of the used parameter values of $A_{ex}$, $Z_{box}$, $V_{total}$?

**Reply:** The comment is accepted, and the terms were elaborated. lines 267-269:

" The total volume ($V_{total}$) was calculated by multiplying the penetration area of the experiment, $A_{ex}$ (m$^2$) =240m$^2$ (see section 2.2) by the depth of the aquifer section used in the model, $Z_{box}$ (m) =10m (determined as the boundary of the model); "

**12.** P. 8, l. 221: I think you mean "The initial mass of perchlorate is…".

 **Reply:** The comment is accepted, and the terms were changed. Line 271: " The initial mass of perchlorate is…"

**13.** P. 8, Eqs. 7 and 8: The unit conversion factor must be "1000 L / 1 m³" for consistency.

**Reply:** the other reviewer asks to delete the unit conversion from the equation. Therefore, Eqs 10-11 were revised.

**14**. P. 9, l. 239: Can you here please give a short explanation or suggestion why the diffusivity in the last two layers is so high compared to the other layers above?

**Reply:** According to the sensitivity analysis that was implemented in the revised manuscript. It appears that the λ parameters were not sensitive. Therefore, the default value applied by the Hydrus code was kept (i.e., 10 cm; Tables 1 and 2).

**15.** P. 10, l. 263: Why even showing and using the observed values in 17 m depth for simulation, when you think that this sampling point does not generate reliable measurements and not capturing the real conditions in this depth?

**Reply:** The vadose zone monitoring system in our research field provided information on 40m of unsaturated zone, and as in any unsaturated zone the level of homogeneity and uncertainty is hard to estimate.  Nevertheless, we presented the results in order to reflect the heterogeneity of the cross-section and the difficulty in evaluating the flow and solutes transport process.  It is important to clarify that the data from 17m depth was not used for calibrating the model but to assess the fitting of the calibrated model to the observation in the field. And for the scientific integrity we decided to keep those unperfect results.

**16.** P. 10, Table 1: The unit of λ is missing.

**Reply:** Table 1 has been revised. The λ parameter is in cm.

**17**. P. 12, l. 310-313: This explanation is not completely clear to me. Can you please provide some more information on this modelling procedure in two phases? How long are the respective phases?

And how do you subdivide Layer 1 from Table 1 (0-2m) into the three sublayers of Table 3 between 0-40 m?

**Reply:** We apologize for the typing mistake in Table 3, the depth of the third layer should be between 1.3 and 2 m thickness. Table 3 was revised accordingly. Furthermore, we added more information regarding the modelling procedure in two phases (lines 387,389-391). Note that the first phase was running for 55 days, and the second phase extended over 233 days.

**18.** P. 13, Table 3: Do you assume $\mu$ as the first-order degradation rate coefficient for the water phase? Thus, do you only assume degradation taking place in the water phase, as you do not say anything about parameterizing sorption? Does perchlorate not adsorb at all to soil particles?

**Reply:** Perchlorate does not adsorb to soil particles as mentioned in line 37: " Perchlorate's high solubility and low sorptivity to soil particles facilitate its distribution in the subsurface "

**19.** P. 14, l. 363: I cannot find a depth of 0.05 m in Figure 4. Please revise

**Reply:** The comment is accepted and we removed reference to the figure.

**20.** P. 15, Figure 7: Labels of x-axis are missing.

**Reply:** Figure 7 has been revised accordingly**.**

**21**. P. 18, l. 446: What does "minimal costs" actually mean? What would a continuous treatment over 900 days cost and what is the difference to common methods

**Reply:**

Compared to different common methods, which usually based on ex-situ treatments (such as external reactors, soil excavation etc.), submersible pump and basic irrigation equipment are the only technical requirement for our proposed treatment. We clarified this point in lines 533-536:

" All ex-situ approaches require relatively high costs and complicated operations during the transport of the contaminants and the establishment of the treatment facilities. Unlike those methods, the proposed approach treats the contamination in-situly and simultaneously for the soil, the unsaturated zone and the groundwater with extremely high perchlorate concentrations in a relatively simple and affordable manner."

---

## Author Comment (AC2)

This interesting manuscript follows a relatively large scale experiment in which contaminated water (perchlorate) is pumped, and the shallow subsurface acts as a reactor for the remediation. The presented work include experimenta work (some of it already presented, but its extent is not clear from the presentation), and a numerical model using the HYDRUS platform. Overal the manuscript is easy to follow, presents an interesting approach (even if somewhat questionable), and is of value. Most of the conclusions make sense. However. too many aspects need to be improved before the manuscript can be accepted for publication. Primarily, the authrs need to convince that their model is reliable - I was not convinced and therefore I could not evaluate the model results and the conclusions drawn by these results

**1.** While the idea is certainly interesting, one should wonder why use the shallow vadose zone, where it is far from being trivial to control pH and oxygen levels, instead using a controlled reactor and let cleane water percolate. After all, the water is pumped anyway. The authors should at the very least to discuss their alternative vs. more cassic pump & treat approach.

**Reply to comment 1:** The concept of using soil as a bioreactor is primarily aimed at reducing costs and eliminating the need for large, expensive, and complex reactors. Soil naturally contains the bacteria necessary for the process, and pH levels can be monitored and controlled through the addition of buffer solutions. Oxygen penetration can also be regulated by covering the area with polyethylene sheets. These two factors - pH and oxygen - can be managed in a relatively straightforward way, as opposed to the complexities involved in constructing new bioreactors.

In addition, we revised the introduction with an overview of more classic approaches for perchlorate treatment. (lines 55-65).

**2.** overall the text is "slopy", especially the introduction and materials and methods - see too many examples below - and should be polished.

**Reply to comment 2:** The introduction was revised accordingly. See the examples below.

**3.** acronyms should be defined, even if the authors believe they are common (they are not).

**Reply to comment 3:** The comment is accepted. All acronyms were defined. See lines 9,16,18,92,122.

**4.** L19 and on This part of the abstract lacks clarity. 70 days or additional 200 days?

**Reply to comment 4:** The comment is accepted, and the abstract was revise accordingly. Line 21-22: " According to modeling simulations, in order to achieve complete removal of contaminants from the groundwater as well, the implementation of the in-situ bioremediation should be continued for an additional 200 days."

**5.** L33 There are so many "conventional methods", that accuracy s needed. The most conventional method is pump and treat, but it is not trivial to call it in-situ method.

**Reply to comment 5:** The comment is accepted, the sentence was revised. Line 32-33:
" Among the in-situ treatment approaches, bioremediation is becoming an increasingly popular alternative."

**6.** L45 if the final product is water, the chemical equations should contain hydrogen and its source (and the consequence of its use - pH change) should be discussed.

**Reply to comment 6:** As mentions in the text: the process continued and the oxygen is also reduced. This part is not presented in an equation (not relevant). Nevertheless, the consequences of the pH was added, lines 51-53: " Several factors can affect the pH level, such as the mineralization of ethanol, oxygen reduction, and different competitive electron acceptors. Previous conclusions have shown the acidification process during the treatment (Levakov et al., 2019) which required buffering and frequent monitoring. "

**7**. L59 what further treatment? If further treatment is needed, why bother with in-situ remediation?

**Reply to comment 7:** The full sentences: "For deep unsaturated zone environments, clean water is injected to displace the pollution to the groundwater from inaccessible layers (Evans et al., 2011; L. Liu et al., 2018; Luciano et al., 2013). Subsequently, the groundwater is pumped for further treatment (Guo et al., 2013; Høisæter et al., 2021)."

As mentioned, further treatment is refer to common treatment from the literature, aiming to solve deep unsaturated zone contamination. In this method, clean water is flushed the deep pore water into the groundwater, and then the groundwater is treated ex-situly (pump and treat). The flushing process is necessary in order to extract the contamination from the unsaturated zone.

We try to clarify this point, lines 67-68: " Usually, the subsequent treatment includes ex-situ methods outside the contaminated site (Guo et al., 2013; Høisæter et al., 2021)."

**8.** L62 by using "the contaminated site" the authors assume that the reader is familiar with the site, which is questionable

**Reply to comment 8:** the introduction was revised. Lines 62-64

As opposed to those methods, the current research site is characterized by a deep unsaturated cross-section (40m) while the major mass of perchlorate is located in the deep layers (17-36 m) (Dahan et al., 2017; Gal et al., 2009; Levakov et al., 2019).

Next, the research area is well described in the methods chapter. Lines 86-98.

**9.** L85 the more appropriate term would be variably saturated zone. e.g., it is more than likely that the clay layer (or the soil just above it) gets saturated at leas occasionally

**Reply to comment 9:** the term unsaturated zone is appropriate and acceptable. No evidence for saturation was observed.

**10.** L119for how long?

**Reply to comment 10:** the methods chapter was revised. Line 130: "(up to 10 days)"

**11.** L129 m^3 or m^-3? not clear

**Reply to comment 11:** the text was corrected. Line 146: "$5 \text{ m}^3$ "

**12.** L129 and elsewhere the term injection is used throughout the manuscript. Is that the right term? my understanding of the system is that it is mostly gravity driven

**Reply to comment 12:** injection is the right term since the water from the storage tank is being injected using a booster pump (in order to increase the pressure in the drip irrigation system).

**13.** L140 raise power

**Reply to comment 13:** we correct the units. Line 155: $0.1 \text{ mg l}^{-1}$

**14.** L163 is the use of first order reaction, for reaction that (as the authors claim) depends on the lack of oxygen and the availability of oxygen donor, justified? Can the authrs show that the oxygen levels, pH and ORP are in the desired ranges. One can speculate that the experiment is simply dilluting the contaminants (to be

clear, I do believe that the experiment is performing as planned, but the text does not support it well neither by measurements (many of which unreliable, as the authors indcate, nor by the model that lacks the complexity and the supprting measurements

**Reply to comment 14:** The comment is accepted and an analysis of oxygen level and pH were added to the supporting information. (see supporting information). Reference in the manuscript:  line 367-368: "(pH values in the supporting information-Figure S4)"

Lines 401-402: "Oxygen levels were measured during the experiment using a biogas analyzer. According to the measurements, no evidence for oxygen was observed below a depth of 0.5m during the degradation state."

In addition, a general clarification on the degradation process was added to the discussion in order to clarify the speculation of dilution instead of degradation. Lines 379-383:

Previous laboratory and field experiments dealing with the current contaminated site have proven that the decrease in perchlorate concentrations is attributed to microbial degradation in the soil (Avishai et al., 2017; Levakov et al., 2021, 2019; Itamar Sikron, 2013). Those experiments examined the assumption using measurements of different products along the process, changes in the microbial population abundance and relevant genes, and organic materials mineralization.

**15.** L167 a word about model heterogeneity would be in place. The subsurface seem to be highly heterogeneous (Fig. 1) and therefore some heterogeneity, physical (VG parameters, porosity) and chemical (reaction rates) should be considered.

**Reply to comment 15:** We agree with the reviewer's comment regarding the vertical variability of the lithology. Essentially, the decision of the parameterization of the model domain is according to the sensitivity analysis (which now added to the manuscript; lines 217-234) and the calibration process (which now improved and more organized; lines 235-248). Therefore, the prescribed and optimal VGM and reaction rates parameters are shown in the 'Results and discussion' section (Table 1, 2 and 3).

**16.** the use of a one-dimensional model should be justified. Clay layers tend to induce lateral flow

**Reply to comment 16:** no evidence for saturation in the clay layer was observed thus we don't expect lateral flow during the treatment. Moreover, we don't have any information to support a 3D model, and no significant benefit from such a model. Below we provide simulated water content at the at the bottom of layer 3, and close to layer 4 (the clay layer). The saturated water content ($\theta_s$) of layer 3

and layer 4 are 0.37 and 0.38, respectively (Tables 1 and 2 in the manuscript). The simulation results illustrate that only unsaturated conditions prevail during the time of the experiment.

[Figure]

**17.** L185 why atmospheric BC? there is no runoff, true, but there is also no rainfall and almost no evaporation is the site is covered by plastic, as mentioned above

**Reply to comment 17:** We agree with the reviewer's comment that using atmospheric boundary conditions (BC) under non-atmospheric conditions might be misleading and worng. However, these boundary conditions were used for technical reasons, since it is easier to implement fluxes and concentrations (zero or non-zero). The default of Hydrus is to switch between flux and head (i.e., $h_{crit}$ is a value that is predifinied). We provided this explanation in the text (lines 203-204).

**18.** L186 Where is the lower boundary condition? still in the vadose zone or in groundwater (and why so)?

**Reply to comment 18:** The lower BC at 40 m depth is prescribed as constant head since water table fluctuations are small (lines 204-205)

**19.** L190 again. those are HYDRUS terms. What are the BCs? For the contaminant, the concentration changes over time - making the BC non-linear and state dependent. How was that taken into account?

**Reply to comment 19:** For the upper boundary condition, a third-type (Cauchy) boundary was imposed. The main advantage of this BC is that there is a full control over the mass balance and how much solute enters into the transport domain by prescribing the solute flux. In the lower boundary we imposed zero concentration gradient (a second-type boundary condition; Neumann type). We added this information to the text (lines 209-210).

**20.** L198 trial and error is fine, but still - some 15-20 parameters are involved. Any details about the process and its convergence would be nice

**Reply to comment 20:** Both reviewers showed concern regarding the application of the trial and error approach for such large parameter space. Therefore we improve our calibration method by first applying a sensetvity analysis to reduce the number of parameters that go through calibration. Subsequently, a calibration method that includes an uncertenty analysis was implemented (lines 214-250).

**21.** L217 it is really funny to see equations written in formal mathematical notation, and then others that make use of asterisk … or others that include unit conversion as part of the equation

**Reply to comment 21**: all the equations were fixed.

**22.** Fig. 3 clearly the model calibration is problematic beyond 2 m. Is that an issue? The authors mostly bame the measurements, possibly rightfully, but they do not discuss the concequence of a less-calibrated model. Are the model results and the conclusions drawn reliable?

**Reply to comment 22:** We acknowledge the reviewer's comment and elaborate the discussion concerning the modeling results (lines 307-316). The calibrated water flow and bromide transport succeeded in simulating the first arrival time and peak bromide concentrations for most depths. However, the model simulations display a longer tailing compared to bromide observations. Note that the objective of the modelling approach is to predict the time which requires to operate the treatment method in order to assess accurately the operation requirements. Furthermore, the treatment time scale is expected to extend over number of years (900 days). Thus, the error in decrease of bromide concentrations is acceptable. We added this clarification in lines 525-528:

"It is important to mention that modelling the unsaturated zone can present significant challenges due to the high complexity of the vadose zone and multiple variables that require for the calculation. Nevertheless, the primary objective of the model is predicting the required treatment duration and therefore minor deviations from the ultimate outcome may be acceptable"

**23.** L263 an alternative explanation may be that the water bypasses this region, that is beneath the clay. Here the question of 1D vs 3D comes to mnd

**Reply to comment 23:** Obviously preferential flow in the unsaturated zone is expected. Nevertheles, the vadose zone monitoring technology that was implenmented in this study enables measures water flow and solute transport in multiple points located under independent profiles of the unsaturated zone (practicaly multiple 1D profiles). Accordingly the Herogenety is also expreseed trough the data. Nevertheles except of the isolated point in depth of 17 m all the other 14 measurement point enabled us to assume that the general

flow direction is vertical and the degree of heterogenety is "small enough"to enable a 1D moel. Moreover we don't have any information to support a 3D model and no significant benefit from such a model (see comment 22 – the aim of the model).

**24.** Fif. 4 for 17 m the model is really off. For 36 m the model captures the general trend, but not the details. Any reason other than easurements?

**Reply to comment 24:** Following the implementation of the uncertainty approach, it seems that most of the observed perchlorate observations at 36 m depth fall within the uncertainty boundaries (new figure 5). Therefore, the model captures both the general trend and the perchlorate concentrations. The observed perchlorate concentrations at 17m depth were presented in order to illustrate the challenges in acknowledging the spatial heterogeneity of texture and concentrations in models Yet we believe that this point is relatively isolated and do not wel present the general flow process and transport across this domain. As was flagged by both reviewers, the number of parameters or the parameter space of the suggested 1D model is vary large. The parameterization of a 3D model requires even larger space. Ultimately, the heterogeneity can challenge the in-situ cleaning approach, and therefore both Continuous monitoring and simulating the expected results are necessary for control purposes.

**25.** location of tables should be re-thought

**Reply to comment 25:** table #1 was relocated, Line 317.

**26.** Table 1 quality of the calibration, per layer, would be nice here

**Reply to comment 26:** We now have implemented a sensitivity analysis and a calibration approach that includes uncertainty estimations (lines 235-250). Note that Table 1 has been modified.

**27.** Fig. 6 I do not see this figure as being useful by any means

**Reply to comment 27:** Figure 6 has been removed.

**28.** Fig 7 horizontal axes is missing

**Reply to comment 28:** Figure 7 has been revised accordingly.

**29.** L391 other than nitrate (that is related in a way to the degradation of the main contaminant), the value of the other co-contaminants is not clear. For nitrate, it ill be useful if the authors can show that its degradation is actually related, and not just assumed so

**Reply to comment 29:** the degradation of all three co-contaminants is important to the rehabilitation of the soil as often Percholrate is produced and dispoed with other explosives. The degradation is shown in figure 8 and disgusted in chapter 3.4. for example, nitrate, lines 477-487:

" In addition, denitrification was observed in the reactive shallow soil, which decreased the nitrate concentration to below the detection limit in the upper 1.5 m (**Error! Reference source not found.**). Several perchlorate-reducing bacteria were found to be capable of using nitrate as electron acceptors (Youngblut et al., 2016). The process of denitrification is considered thermodynamically preferrable to (per)chlorate reduction (Xu et al., 2003) and, therefore, was observed sooner. Nitrite, as a by-product of the process of denitrification, was not detected throughout the experiment at any depth (data not shown). The increase of chlorate and nitrate at depths of 2.6 to 13 m was due to the initial application of the polluted groundwater when the microbial population was not yet adapted. The initial pollution front that progressed along the unsaturated zone was partially degraded by the developed microbial populations until 13 m, where no evidence of chlorate and nitrate was observed due to biodegradation and dilution. During the rest of the experiment, while applying high concentrations of chlorate and nitrate to the shallow soil layers, continuous bioreduction was responsible for maintaining the low concentrations of the co-contaminants."

Nevertheless, we clarified the point in lines 487-489:

Evidence for conjugated nitrate and perchlorate degradation was observed in different in-situ treatment sites (Evans et al., 2011b; Höhener and Ponsin, 2014; Lorah et al., 2022; Zhao et al., 2022). Moreover, the correlated trends of those two components can also demonstrate the mutual source of the processes.

---

## Author Comment (AC3)

Dear prof. Zehe

First, we would like to express our sincere gratitude for your input and expertise in refining the manuscript. Your constructive feedback has contributed to improving the quality and impact of the work.

Our choice of using one-dimensional modeling is due to the lack of evidence or observations for two and three-dimensional processes such as lateral flow. The VMS was installed diagonally in order to capture the vertical flow in undisturbed profiles. Accordingly, all monitoring points are shifted vertically and laterally a few meters apart from each other. Nevertheless, data from these multiple 1D vertical profiles cannot provide information for such multi-dimensional processes. The implementation of a more complicated model must be backed up with more 3D geological and hydrological information, which currently is lacking.

Regarding the bromide correlation, results from 2.6m didn't match the model prediction. Nevertheless, both peaks at 5.5 and 2.6m depth were completely missing from the observations. We assumed they were not measured due to the large gap in sampling during that period (3/2016). No other significant peaks were observed to provide information for preferential flow. We elaborated on those results in lines 307-316.

Both reviewers showed concern regarding the choice of modeling methods. Specifically, the issue of parameters' sensitivity and optimization, due to the large number of parameters involved. Therefore, we implemented the Morris method sensitivity analysis (Morris, 1991) using the SAFE Matlab code provided by Pianosi et al. (2015). In this method, the parameters are modified one at a time and sensitivity is estimated as the partial derivative of the change in model output for a given change in a single input parameter (Perzan et al., 2021). Subsequently, the sensitive parameters were calibrated following the method presented by Perzan et al. (2021). As part of the optimization procedure, the uncertainty and optimal values are calculated according to the behavioral simulations (by achieving evaluation goals).

In addition to the model improvements, several adjustments were added to the manuscript according to the reviewers' comments such as the elaboration of different treatment approaches around the world, the addition of soil parameters to the supporting information (oxygen and pH value in the soil), and further specific clarifications.

Ultimately, simulating the water flow and reactive transport in the unsaturated zone presents significant challenges due to the high complexity of the vadose zone and the multiple variables that are required for the calculations. However, the main purpose of the model is to forecast the duration of treatment needed, and thus slight deviations from the desired outcome may be acceptable. The available data sets that were obtained in the current field experiment do not support a 3D model. Thus, implementing such a complicated model would not be beneficial.

Morris, M.D., 1991. Factorial sampling plans for preliminary computational experiments. Technometrics 33, 161–174.

Perzan, Z., Babey, T., Caers, J., Bargar, J.R., Maher, K., 2021. Local and Global Sensitivity Analysis of a Reactive Transport Model Simulating Floodplain Redox Cycling. Water Resour. Res. 57, 1–24.

Pianosi, F., Sarrazin, F., Wagener, T., 2015. A Matlab toolbox for Global Sensitivity Analysis. Environ. Model. Softw. 70, 80–85